# Improvements in the Effectiveness of Early Detection in Colorectal Cancer with Open-Label Randomised Study

**DOI:** 10.3390/jcm10215072

**Published:** 2021-10-29

**Authors:** A. Álvarez-Delgado, M. L. Pérez García, J. M. García-González, H. Iglesias de Sena, A. J. Chamorro, M. F. Lorenzo Gómez, M. Marcos, J. A. Mirón-Canelo

**Affiliations:** 1Digestive, Department of Medicine, University Hospital of Salamanca-IBSAL, University of Salamanca, 37007 Salamanca, Spain; albertoalvdel@yahoo.com; 2Institute of Biomedical Research of Salamanca (IBSAL), 37007 Salamanca, Spain; u112076@usal.es (M.L.P.G.); ajchamorro@gmail.com (A.J.C.); 3Internal Medicine, Department of Medicine, University Hospital of Salamanca-IBSAL, University of Salamanca, 37007 Salamanca, Spain; 4Department of Sociology, University Pablo of Olavide (Sevilla), 41013 Sevilla, Spain; jmgargon@upo.es; 5Department of Biomedical and Diagnostic Sciences, School of Medicine, University of Salamanca, 37007 Salamanca, Spain; hidesena@usal.es; 6Department of Urology, University Hospital of Salamanca, 37007 Salamanca, Spain; mflorenzogo@yahoo.es

**Keywords:** colon cancer, population-based screening, intervention group with a control group, effectiveness, quality of the intervention

## Abstract

Introduction: The general objective of this research is to improve the quality of colorectal cancer screening (CRC) by assessing, as an indicator of effectiveness, the ability of colonoscopy to detect more advanced adenomas in the exposed group than in the control group. Material and Methods: The present work is designed as an open-label randomized study on cancer screening, using two groups based on their exposure to the protocol: an exposed to intervention group (EIG, 167), and a control group (CG, 167), without the intervention of the protocol and by 1:1 matching. Results: In 167 patients in the GEI, 449 polyps are visualized and 274 are adenomas (80.58%), of which 100 (36.49%) are advanced adenomas. In the CG (*n* = 174), there are 321 polyps and 152 adenomas (82.60%). The variables significantly associated by logistic regression to the detection of adenomas are the male sex with an OR of 2.52. The variable time to withdrawal, ≥9 min, is significant at 99% confidence (*p* = 0.002/OR 34.67) and the fractional dose is significant at 99% (*p* = 0.009, OR 7.81). Conclusion: Based on the observations made, our study suggests that the intervention in collaboration between primary care and hospital care is effective from a preventive point of view and achieves the objective of effectiveness and quality of the PCCR.

## 1. Introduction

Colorectal cancer (CRC) has become one of the most frequent neoplasms worldwide in the last few decades, with few differences between men and women [1]. Colorectal cancer is predominant among elderly people with a mean age of 70–71 years, and a large percentage of patients are diagnosed after the age of 50. Regarding sex, CRC affects men and women almost equally [2]. According to Globocan estimates, the most frequently diagnosed cancers in Spain in 2020 are rectal and colon (44,231 new cases), prostate (35,126), breast (32,952), lung (29,638), and urinary bladder (22,350). The Spanish Association Against Cancer reports that colorectal cancer is the most prevalent when considering both genders, and is the tumour with the second-highest mortality in Spain [3]. At present, colorectal cancer is the most frequently diagnosed neoplasm in Spain (representing 15% of all tumours) and the second cause of death due to cancer in our country. Over 37,000 people are diagnosed with colorectal cancer in Spain, the highest incidence of all types of cancer if we account for both sexes, and is the second cause of death by cancer with approximately 15,000 deaths every year [4]. Worldwide, colorectal cancer is the third most frequently diagnosed malignancy and the second cause of cancer death. It has a high incidence and its association with mortality at a global level ranks this disease in third place in incidence and second in cancer mortality, according to the Globocan database [5,6]. In countries with high incidence, mortality has stabilized or even diminished, reaching 50% survival rates, probably due to the use of techniques such as the endoscopic polypectomy, especially considering that the stage of the tumour at the time of diagnosis is the main prognostic factor [7,8,9].

Population-based CRC screening is part of the early detection strategy for cancer of the Spanish National Health System, following European recommendations [10]. In Spain, a few colorectal cancer-screening initiatives were developed in the 1990s in small populations. The 2005 Spanish National Health System’s Cancer Strategy recommended the implementation of pilot colorectal cancer screening programs. In an update in 2009, the strategy proposes progressive implantation of the program and sets a target coverage rate of 50% by the year 2015 [11].

The screening and early diagnosis included in secondary prevention in CRC include several early detection strategies available with proven effectiveness and efficiency. The most recommended tests used are the faecal occult blood test in countries with population screening programs, because of its high sensitivity. A secondary option is the colonoscopy, because of its high sensitivity and specificity; this is the diagnostic and therapeutic test that is the culmination of strategies used in institutions [12,13].

The European Commission edited a quality assurance guide for colorectal cancer screening and diagnosis, which sums up available evidence and establishes recommendations on various methodological aspects [11,14]. Despite the strong evidence supporting the convenience of undertaking population-based screenings for this type of cancer (these programs are relatively well implemented in Europe), they only cover 43% of the target population, and there is still some variability as to the type of screening test used [15].

The definitive diagnosis of CRC comes from histological confirmation after the colonoscopy. At present, a complete colonoscopy is considered critical when ruling out synchronic lesions present in around 2–4% of cases and, should a complete colonoscopy be unavailable, colonography using computerized tomography (CT) can help visualize the rest of the colon [16]. Around 20% of recently diagnosed CRC patients show distant metastasis, the liver being the most frequent location. In an extensive CRC registry in France, pulmonary metastatic repercussions are found in around 2.1% of cases [17], and are three times more frequent in rectal cancer specifically, which would justify the use of thorax CT scans in patients with locally advanced cancer.

A better understanding of the natural history and the factors associated with CRC allows for the implementation of prevention programs intended to avoid its occurrence (primary prevention), detect it in its early stages (secondary prevention), or improve its prognosis after onset (tertiary prevention) [18,19].

Screening programs, conceived as tools to minimize the impact of cancer on a population, justify the need for organized actions to ensure better coverage for the population, including the more vulnerable and/or hard to reach groups. Population-based screening is more effective and cost-effective than opportunistic screening. Evidence indicates that an organized screening program achieves greater reductions in incidence and mortality for CRC [20].

The effectiveness of the screening program will depend on multiple factors, such as accessibility and acceptance by the population, and the efforts of healthcare professionals. Screening programs should reach as many people as possible, seeing as they are known to be effective and cost-effective, even with coverage rates below 40% [19,20]. Furthermore, CRC screening has one of the best cost-effectiveness rates of all screening and other preventive and diagnostic practices, which is why it is one of the priority recommendations of scientific societies to health professionals [12,13]. We are aware that the effectiveness of screening will depend on multiple factors, such as accessibility, patient acceptance, and preparation and, specifically, on the quality of care of the health professionals.

The aim is to analyse the protocol designed and implemented in a gastroenterology department as a screening program, which can improve the diagnostic efficacy of adenoma and advanced adenoma.

## 2. Materials and Methods

The design used in the present study is an experimental or intervention study in CRC screening, comparing two groups, one exposed to an intervention protocol (GEI) and the other a control group (GC), without intervention. This type of work falls within the framework of clinical preventive research [21,22]. The scope of this study is circumscribed to the Salamanca Health Area, which, according to data from the Regional Government of Castilla y León, has a reference population of 335,986 people covered by 36 primary care teams that derive their patients from the Salamanca Clinical University Hospital (HCU).

This study was conducted in the context of the CRC Detection Program (CRCDP) of the Regional Government of Castilla y León, in which all patients between the ages of 50 and 69 years receive a letter with information and an invitation to participate in the CRCDP. Organized cancer screening tests offered to healthy individuals must meet certain prerequisites, essentially, whether they have been proven to reduce general mortality, disease-specific mortality, and the incidence of advanced disease, and if their benefits and risks are well known and the risk/benefit ratio is acceptable [21]. The inclusion criteria are patients between 65 and 69 years old, who agree to participate with informed consent, are interviewed in person at their primary care centre of reference by the practice nurse and/or family physician from the program and, if eligibility criteria are met, are handed an FOBT test (quantitative faecal immunochemical test/OC-SENSOR^®^ (Eiken, Japan), with a cut-off point of 100 ng/mL and a single sample). Additionally, they are given information, advice, and recommendations for colon preparation (plus the information sheet). The informed consent for the colonoscopy is also addressed at this time (Ethical principle of Patient Autonomy). The nurse provides patients with sachets for bowel preparation (polyethylene glycol 4000, casen-glicol^®^, 4 L—PEG 4 L), while the physician confirms and validates the informed consent. As required, a request is sent to the Clinical Ethics Committee of the Salamanca Clinical University Hospital for approval.

The patients in the study number 334,167, and are randomly selected for the experimental and intervention group (EIG) with sequential sampling. A total of 167 for the control group (CG) are randomly selected with random number sampling. All patients selected from the target population of the screening program are between 65 and 69 years old. The sample is collected, using phone calls rather than face-to-face interviews, as the patients who were identified from screening programs came from primary care. The study design is an open-label randomized trial in the intervention group (GEI) and control group (GC).

***Intervention Group (GEI)****:* Patients were randomly selected by sequential sampling and scheduled for colonoscopy (CDCRP) by the HCU appointment service based on a protocol. Patients were included randomly and consecutively, and all those patients cited since 1 January 2019 were included.

***Control Group (CG)****:* Patients selected as controls were randomly scheduled by random numbers through the HCU appointment service to undergo colonoscopy concerning CRCPD. Their inclusion was consecutive among those cited from 1 March 2017 to 30 June 2018. The performance was according to standard clinical practice.

1.A Summary of the Prevention and Clinical Protocol for the IG is Found Below:

Phase 1: Informative session for physicians from the Clinical Gastroenterology Department on the importance of quality colonoscopy procedures, with special attention to quality indicators, the risk of interval cancer, the quality of the polypectomy, and patient sSafety.

Phase 2: An individual and personal phone call to each patient between 5 and 14 days before the colonoscopy, underscoring the key elements of the procedure. The nursing staff see the patient upon arrival to run a verification survey before placing a peripheral venous catheter.

Phase 3: The colonoscopy starts under deep sedation using propofol ± opioids, monitored by intensive care specialists during afternoon procedures and by the endoscopists themselves during morning procedures. Colonoscopies are conducted using high-definition endoscopes (Olympus, video-colonoscope CF-H190L) and Narrow-Band-Imaging (NBI) optical/digital chromo-endoscopy. All sedation-related complications and drug dosages are recorded accordingly.

Phase 4: Bowel cleansing is evaluated using the Boston Bowel Preparation Scale (BBPS) in three segments of the right (caecum and ascending), transverse and left (descending, sigmoid, rectum) colon, using water irrigation pumps through the channel of the endoscope to remove any residual stool. The global score of the scale is 10 points (0–9). The scale for each segment is of 4 points (0: unprepared colon segment, 1: residual stool and/or liquids, 2: minor residual staining, 3: no residual staining). Cleanliness was evaluated globally and by segments. A global score below 5 indicates deficient preparation. Colonoscopies with very deficient preparations will be suspended and rescheduled for another day. A score of 8–9 indicates very good or excellent preparation, a score of 6–7 is indicative of good preparation. Three degrees of the bowel cleanliness variable were established, scores 8–9 (BBPS 3), scores 6–7 (BBPS 2), and scores ≤ 5 (BBPS 1).

Phase 5: During each exploration, confirmation of caecal intubation is evaluated, and the caecum is documented through endoscopic imaging. If caecal intubation is not achieved, the reason why will be recorded accordingly. When measuring the rate of caecal intubation, the cases in which it was not possible due to benign or malignant stenosis will be included, but not when secondary to severe colitis or deficient preparation. The withdrawal time of the endoscope from caecum to extraction was measured and was classified into three categories, ≤6 min, 7–8 min, or ≥9 min.

Phase 6: All visible polyps were documented and characterized based on the Paris classification (morphology), size measured in centimetres, location by segment or distance to the anal margin, NICE classification (based on NBI), digital chromo-endoscopy or with stains to visualize the margins and surfaces in the case of flat polyps, and endoscopic dyeing in potentially malignant polyps or tumours. Endoscopists decided on the use of cold or hot snares or biopsy forceps during polypectomies on a case-by-case basis.

Phase 7: The complications of the colonoscopy, mainly perforation or haemorrhage, were recorded. Acute or delayed bleeding post-polypectomy was treated according to endoscopic standards. All the removed polyps underwent histological analysis. After the colonoscopy, patients were supervised by nursing staff at the recovery area. Finally, patients were evaluated using the modified Aldrete scale, the results of which were registered accordingly, and were discharged if they scored ≥9 points.

2.Sample

The patients in the study numbered 334,167, and were randomly selected for the experimental and intervention group (EIG) with sequential sampling. A total of 167 from the control group (CG) were randomly selected with random number sampling. All patients selected from the target population of the screening program were between 65 and 69 years old. The sample was collected as an open-label trial, using phone calls rather than face-to-face interviews, as the patients who were identified from screening programs came from primary care.

3.Variables Studied

The studied variables were sex, comorbidities, use of antiplatelet drugs, use of anticoagulants, allergies, the patient’s initiative to participate, patient’s preparation, food intake before the intervention, withdrawal time, tolerance to the preparation, bowel cleanliness with Boston scale, caecal intubation, split dosing, clinical-preventive relevance, effectiveness of colonoscopy as a treatment, and location of the adenomas.

4.Statistical Analysis

The information provided by participants was analysed using the SPSS statistical package version 22.0. Results are presented as frequencies (*n*) and percentages ± their 95% confidence intervals (% ± CI). Additionally, the means ± standard deviations (mean ± SD) were included for quantitative variables. The association between the dependent and independent variables was evaluated using Pearson’s Chi-squared test when the expected frequencies were sufficiently large and Fisher’s exact test when the sample size was too small. The analysis of statistically significant associations was conducted using the Odds Ratios (OR) and Cramer’s V (for 2 × J or I × 2 tables), or Phi (φ) (for tables with dimensions equal to 2 × 2). Finally, a multivariate logistic regression analysis was performed. Statistical significance was set at 0.05 (*p* < 0.05) with a 95% confidence level.

## 3. Results

### 3.1. Characteristics of the Population

There were no significant differences in age, sex, comorbidities, use of antiplatelet drugs—16 (9.58%) in IG and 18 (10.34%) in CG, use of anticoagulants—9 (5.38%) in IG and 11 (6.32%) in CG, and allergies between the intervention group (IG) and control group (CG).

### 3.2. Patient’s Initiative to Participate and Preparation

From the total of 167 patients, 129 (77.24%) answered the initial call and 29 (17.36%) did not. In the GC, no call was made, unlike in the GEI, which was made to improve the preparation and intestinal cleansing. The patients made fractional doses of the preparation in 73.65% of the cases, although 6 patients did not do so.

### 3.3. Food Intake before the Intervention, Withdrawal Time, and Tolerance to the Preparation

The differences between the IC and CG in food intake before the intervention: fasting time was predominantly <6 h in 79.65% (133/167) of cases, with 4–6 h fast times being the most frequent in 49.7% (83/167) of cases, while 20.35% (34/167) fasted for >6 h. Patients prepared correctly in 77.84% (140/167) of cases. Out of the patients who received the call, 30 (23.25%) did not prepare correctly, mainly because the fasting period before the procedure was <6 h. Bad tolerance (nausea, vomiting) to the preparation occurred in 9.58% of cases, all using PEG 4 L.

Bowel cleanliness data are presented in Table 1. Patients in the IG showed better preparation measured using the Boston Scale, both in the global mean score 7.71 and in the percentage of patients with good or excellent preparation—93.62% in the IG, compared to 87.89% in the CG. Bowel cleanliness by segments in the IG was good or excellent in over 90% of patients in all segments. Cleanliness in the right colon was bad or deficient in only 11 patients (6.6%).

### 3.4. Effectiveness of Colonoscopy as Treatment

In the clinical-preventive relevance and effectiveness of colonoscopy, 449 polyps were visualized (one case with polyposis) out of the total of 167 examinations performed in the GEI. The nine polyps that were not removed were polyps <5 mm in the rectum, suggestive of hyperplasia by NBI, according to the endoscopist’s criteria. The total number of adenomas was 274 (80.58%), of which 100 (36.49%) were advanced adenomas.

In the GC of 167, 321 polyps were found and 310 polypectomies were performed; in this case, 11 polyps were not removed because they were of a small size, located in the rectum, and/or had a hyperplastic appearance. Out of the total number of polypectomies, 127 lesions (39.56%) were not recovered for histological study. The total number of adenomas was 152 (82.60%), of which 77 (50.65%) were advanced adenomas (see Figure 1).

### 3.5. Adenoma Location

The most frequent location of the adenomas that were removed in the study group was distal to the splenic flexure, and numbered 144 (52.55%). By segments, the most frequent location in the IG was the sigmoid colon, and numbered 74 (27%). Eighty-seven (31.74%) adenomas were found in the right colon and caecum. In the CG, 42 (27.63) adenomas from the right colon and caecum were analysed. The most frequent location was the sigmoid colon, and numbered 45 (29.6%) (Table 2).

In the IG, adenomas (274), adenomas in advanced stages (100), and adenomas with ADC pt1 (8), were diagnosed. More adenocarcinomas (14/167) were diagnosed in the CG. After adding up ADC pt1 and ADC, the results are practically equal for both groups in number and percentage—IG 14/167 (8.4%) and CG 13/167 (7.8%).

At least one lesion compatible with an adenoma was removed in 121 of the 167 colonoscopies conducted in the IG, meaning the rate in adenoma detection (ADR) was 72.45%, and in colonoscopies in the CG (49.7%), *p* = 0.003. The high-risk group advanced adenoma detection rate (aADR) was defined by the presence of more than three adenomas, an adenoma larger than 10 mm, an adenoma with a villous histological component (>20%), or an adenoma with high-grade dysplasia/ADC in situ. At least one polyp was removed in 99 of the 167 colonoscopies (56.89%) conducted in the CG (ADR), and in 63 of them (36%), *p* = 0.0016, they were classified as advanced or high-risk adenomas (aADR). Table 3 represents the significance studied and associated variables after bivariate analysis on the detection rate of advanced adenomas.

### 3.6. Complications of the Colonoscopy

There were no reported complications during the colonoscopies.

### 3.7. Logistic Regression

Logistic regression was performed as multivariate analyses; the differences in detection between the IG and CG during colonoscopies were significant in the rate of adenomas (*p* = 0.003 and RR/OR of 1.6 and 1.9) and in the detection rate of advanced adenomas (*p* = 0.016 and RR/OR of 1.25 and 1.7). There was a significant association between the adenoma detection and the variable male, withdrawal time, caecal intubation, the initial call, patient preparation, split dosing, and fasting hours (Table 4).

The male category was significant with a 95% confidence level (*p* = 0.035). The OR of the male category is 2.52. The withdrawal time ≥ 9 min was significant with a 99% confidence level (*p* = 0.002/OR 34.67). The value of 7–8 was not significant (*p* = 0.103). Split dosing was significant with a 99% confidence level (*p* = 0.009, OR 7.81).

## 4. Discussion

CRC cancer is one of the neoplasms that shows some of the greatest benefits from preventive measures, especially screening and early diagnosis included in secondary prevention. For this reason, there are several early detection strategies available with proven effectiveness and efficiency. The most recommended tests used are the faecal occult blood test in countries with population screening programs, because of its high sensitivity and specificity [12,13]. In most countries, population-based CRC screening is performed as opposed to opportunistic screening. Colonoscopy is the diagnostic and therapeutic test that is the culmination of the strategies used in institutional CRC screening programs, so their efficiency depends largely on the effectiveness of patient preparation and the competence of the person performing the colonoscopy. As a preliminary element, patient information is necessary to increase participation, and the evidence indicates that receipt of a letter signed by the family physician increases patient participation and collaboration [23,24].

Regarding the study groups, both the IG and CG were similar and homogenous, and therefore, comparable, even though the protocol was not applied to the CG, making withdrawal times unavailable. This could be one of the limitations of our study, though hardly relevant.

In addition, previous experience demonstrates that bowel cleansing is essential for proper visualization and detection of lesions in the colon because inadequate cleansing is associated with incomplete examinations and increased inefficiency [25,26]. The information and recommendations on a correct bowel cleansing technique are critical to improving results, as was observed in a meta-analysis published in 2015 [27]. The same authors conveyed that this can be achieved both through direct and indirect methods. For this study, an indirect system was chosen to inform patients, using phone calls instead of in-person interviews, since patients from screening programs came from primary care. PEG 4L was used in both groups due to its availability in primary care centres and its high efficacy [26]. Compliance with the ingestion of PEG 4L was 86% in the IG, probably due to split dosing [28,29], and intolerance to the product was low (9.5%), similar to other studies [30], where the procedures were undertaken in the afternoons with a split-dose regimen. Split dosing has demonstrated higher effectiveness than single-dose intake [30]. Fasting is another factor related to cleanliness, and fasts of under 6 h are associated with better cleansing in general. In the right colon, the use of a split-dose PEG for bowel preparation before colonoscopy significantly improved the number of satisfactory bowel preparations, increased patient compliance, and decreased nausea, as compared with the full-dose PEG [31]. In the IEG, patients with fasting <6 h predominated 79.64%, in contrast to the CG, where all were >6 h. Thus, with fractionated doses and with fasting hours <6 h, awareness-raising and telephone education, overall bowel cleansing was better than in the CG (BBPS 7.71 vs. 7.43). In other words, good or excellent preparation was in 93.6% of patients in the IGT and 87.79% in the CG, with statistical significance.

Withdrawal time directly measures the motivation and interest of endoscopists in the revision of all colonic segments and previous observations report that only withdrawal times > 8 min and split dosing are associated with an increase in adenoma detection, and only withdrawal times > 8 min showed higher advanced adenoma detection rates. In this study, withdrawal time was >9 min in 80% of patients [32]. This observation may be because it is a university hospital where there are residents in training and medical students; therefore, they are aware of the need to teach well and to obtain good diagnostic effectiveness as one of the learning objectives by demonstrating that doing things well and spending more time on examination is good for the health and care of patients.

The adenoma detection rate (ADR) is a direct indicator of the effectiveness of colonoscopy screening and is influenced by other factors; however, it is an objective and easily measurable indicator that reflects the dedication, competence, and professionalism of the endoscopist in colonoscopy. It is of diagnostic and clinical significance because it is significantly associated as an independent factor with interval CRC [33,34]. The advisable ADR in occidental populations when the screening strategy includes immunochemical faecal occult blood tests must be over 40% (Grade A recommendation, Level of Evidence 1c) [35]. Our intervention well exceeds this figure, with 72.41% being comparatively high for the existing literature on patients participating in screening programs. These results are conditioned by the fact that the intervention of this research aims to improve efficiency and effectiveness by improving the quality of care for patients who are screened. Furthermore, the results observed in the IG are significantly superior to those in the CG, with an ADR of 56%. These results could be conditioned by the actions of family physicians, who could have included patients willing to have an earlier diagnosis or patients who had already been diagnosed with CRC during screening. Both situations could cause detection rates to be biased, though we consider it unlikely, as they were instructed and advised against these practices beforehand to avoid bias. In addition, the transcendence of ADR has been amply reported [35,36], given its inverse association with the risk of interval colorectal cancer, advanced stage interval cancer, and fatal interval cancer. The independent variables associated with ADR, according to the regression analysis, are the following: male sex, withdrawal time > 9 min, answering the initial phone call, patient preparation, split dosing, fasting period, and caecal intubation.

The aADR expresses an even more transcendent result than adenomas in general, due to the number and size of the lesions and the higher likelihood of advanced presentations. The aADR in the IG was very high (49.1%) compared with the CG (36%), similar to previous experiences reported in a systematic review [37]. A recently published study concluded that an appropriate follow-up using high-quality colonoscopy is crucial for the prevention of CRC in patients with risk factors [38]. The recommendation on CRC screening in medium-risk populations from the Spanish Society of Family and Community Medicine and the Gastroenterology Society is to achieve the implementation of universal screening with the participation of primary care in collaboration with hospital care [39]. This is something we agree with, given that one of the strengths of this study is that we have worked in interdisciplinary teams with primary care physicians.

The independent variables associated with the multivariate analysis are male sex, withdrawal time > 9 min, caecal intubation, answering the initial phone call, patient preparation, split dosing, and fasting period. This implies that these variables are very important both in the detection of adenomas in general and advanced adenomas in CRC.

Additionally, the latest European guidelines on CRC screening quality assurance removed the differentiation between medium and high-risk groups, eliminating a considerable number of early detection colonoscopies. Recommendations against the need for follow-up in the low-risk group were also emitted, and though European guidelines were slightly ambiguous [40], this aspect was not considered in our study since no risk stratification aspect may bias the observations.

In the latest meeting of the American Gastroenterological Association, several clinical studies with the evaluation of different CRC screening strategies as their objective were shown, highlighting the importance of not delaying endoscopic studies after a positive test result, and several studies confirmed the importance of high-quality colonoscopy in CRC detection programs [12,41]. Its implementation has reduced mortality due to proximal lesions and interval cancers, which is the main result of CRC screening, and this is fundamentally associated with the competence and speciality of the endoscopist [42].

Concerning the evaluation of the quality indicators of colonoscopies and their controls, we are aware that the success of screening depends to a large extent on the results obtained by the colonoscopy diagnostic test [43,44]. We have used the rate of polypectomy and the rate of detection of adenomas and advanced adenomas, previously established by other studies, as indicators of quality and effectiveness [45,46].

Our results suggest that the groups studied, both in the IGT and the CG, are similar and homogeneous and, therefore, comparable. The group selection was by randomization based on random numbers and the professionals of the digestive service of the Colonoscopy Unit were unaware of the patients who were being selected and included in the GEI sample, so there was no observer bias, as they performed their work as if the patients were from the population screening. For the control group, this was selected from the screening program register in another period to improve objectivity. In both groups, the members of the unit intervene according to the care program, and all of them, except the MIR residents who are in their learning period, carry out their work in this unit because they are specialists and are trained to perform quality colonoscopies [42,47].

As for a limitation in the CG, as the intervention protocol was not applied, the withdrawal time was not recorded. This may be one of the methodological limitations of the study, but we consider it to be of no great importance. As for the fact that the observations could be conditioned by patients predisposed to the call, we would like to know this aspect to improve the process and its quality through adequate intestinal cleansing. This is important for the development of the colonoscopy with a better preparation due to patient collaboration and sensitization, and the effectiveness of the diagnosis might increase.

In conclusion, based on the observations made in this study of collaboration between primary care (PC) teams and the Digestive Service of the Reference Hospital, our results highly suggest that collaboration between the two levels of care of PC and hospital care is effective from a preventive point of view and achieves the objective of effectiveness and quality of the PCCR. Therefore, the use of intervention protocols to improve the quality of colonoscopies and population screening should involve multidisciplinary teams of healthcare professionals from both levels of care, in order to improve early detection and, consequently, improve the survival of CRC patients.

## Figures and Tables

**Figure 1 jcm-10-05072-f001:**
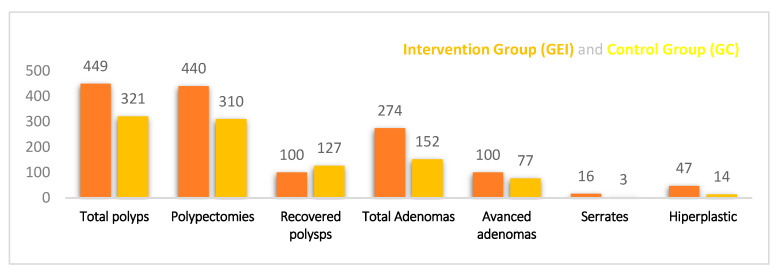
Polyp description in the intervention and control groups.

**Table 1 jcm-10-05072-t001:** Bowl cleanliness using BBPS in both groups.

	EI Group *n* = 167	Control Group *n* = 167
Global BBPS (0–9)	771	743
Global BBPS * (3)	113/165 (68.48%)	84/172 (42.83%)
Global BBPS * (2)	42/165 (25.14%)	67/172 (38.95%)
Global BBPS * (0–1)	8/165 (4.79%)	23/172 (13.37%)

* Boston Bowel Preparation Scale (BBPS): 3 (score of 8–9, very good or excellent). 2 (6–7, good) and 0–1 (<5 bad or deficient).

**Table 2 jcm-10-05072-t002:** Adenoma locations in the IG and CG.

Adenoma Locations	IG (N Adenomas) *	CG (N Adenomas) *
Distal to the splenic flexure	144 (52.55%)	90 (59.2%)
Rectum	24 (8.75%)	18 (11.84%)
Sigmoid colon	74 (27%)	45 (29.6%)
Left colon	46 (16.78%)	30 (19.7%)
Transverse colon	43(15.69%)	20 (13.15%)
Right colon-caecum	87 (31.74%)	42 (27.63%)

* Intervention group (GEI) and control group (CG), number (N).

**Table 3 jcm-10-05072-t003:** Variables associated with the detection of Advanced Adenomas.

			Advanced Adenoma Detection Rate (Yes/No)		
Variables	Cramer’s V/Phi	*p*-Value chi-2	RR1	RR2	OR
Sex (men/women)	0.124 *	0.022	0.804	1.372	0.586
*Endoscopists (1 to 12)*	0.320	0.146			
*Withdrawal time (≤6, 7–8, ≥9)*	0.429 **	0.0001			
Caecal intubation (yes/no)	0.142 **	0.009	0.563	.	.
Call 2 (yes/no)	0.267 **	0.001	2.209	0.430	3.889
*Call 3 (yes/no/NR)*	0.269 **	0.002			
*BBPS 3 cat (deficient or bad/good/excellent)*	0.152 *	0.020			
BBPS 2 cat (deficient-bad/good-excellent)	0.002	0.984			
Split dosing (yes/no)	0.092	0.233			
*Fasting hours (4, 4–6, >6)*	0.119	0.310			

Significance: * 95%; ** 99%; In italic, variables with more than two categories generate tables larger than 2 × 2, for which ORs and RRs cannot be calculated. NR: no response.

**Table 4 jcm-10-05072-t004:** Variables associated with the detection of adenomas.

Variable	Categories	Odds Ratio	*p*-Value
Sex	Women	Ref.	
	Men	2.519 *	0.036
Age		0.997	0.971
Phone call	Yes	Ref.	
	No	12.266 **	0.005
	No answer	10.307 *	0.053
Withdrawal time	≤6	Ref.	
	7–8	4.605	0.193
	≥9	34.668	0.002
BBPS 3 cat	Excellent	Ref.	
	Deficient or bad	0.618	0.553
	Good	2.029	0.203
Fasting hours	<4	Ref.	
	4–6	1.669	0.336
	>6	1.690	0.443
Split dosing	No	Ref	
	Yes	0.128 **	0.009

Method: Variables. Significance: * 95%; ** 99%.

## Data Availability

The manuscript is part of the results obtained in a Doctoral Thesis and these are registered in the Teseo Database of the Ministry of Education (Spain). Available at: https://www.educacion.gob.es/teseo/irGestionarConsulta.do (accessed on 7 September 2021).

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
