# Peer review of "Improvements in the Effectiveness of Early Detection in Colorectal Cancer with Open-Label Randomised Study"

_jcm, 2021, doi:10.3390/jcm10215072_

Round 1

Reviewer 1 Report

There maybe some merit in the paper but very difficult to follow the exact differences between the contool group and the intervention group shaould be clearly outlined It appears the intervention group had a better outcome regarding bowel prep and adenoma detection which is important However the introduction is far to long and covers every aspect of screning rather than focusing on the aim of this study The same applies to the discussion 

Will need major re-editing  and improvent on the written English

Author Response

Estimado revisor:

Tenga en cuenta que he tratado de responder a sus recomendaciones en el manuscrito y que sinceramente les agradezco sus comentarios.

Les adjunto lo que he hecho y lo que podrán ver en el nuevo manuscrito.

Mi agradecimiento por ayudarme a mejorar su calidad.

Atentamente

Reviewer 2 Report

Thank you for this manuscript. It describes an interesting approach in CRC screening, although the main objective of the study is unclear: is the intervention meant to increase participation in screening or the efficacy of screening itself (or just the efficacy of the colonoscopy act) or all of these combined? There are many flaws in the methodology (no clear primary outcome, no sufficient description of the randomization process, information on blinding of GI, no results for harms…) and the studied population is quite small in the end. The limitations of the study do not appear in the discussion, such as the selection bias (only willing participants that do not represent the whole population) or the intervention bias (who is performing the colonoscopy and are they not influenced by the study taking place?) or others. Also, the manuscript needs serious English editing. 

Major revisions :

1/Regarding English language

This article needs extensive editing of the English language. There are many mistakes and complicated sentences.

2/Regarding the abstract

It is unclear when you read the abstract what the intervention/protocol is. It should be described. Also, GIST is not defined

The abstract also needs extensive English editing, many sentences are very hard to understand, especially in the second half.

3/Regarding the Introduction:

It is too long and messy regarding organization of paragraphs. The first paragraph is too long, and then you come back to CRC in the 4th, please be more concise.

It would have been interesting to cite papers that have already studies such interventions (call to patients, incentives…) in CRC screening program.

4/Regarding the Methods

The primary outcome of the study is unclear and not mentioned as such.

You need to explain how randomization was done: “using random numbers” is not enough

Was the study entirely open-lable?

Who are the GI performing the colonoscopies ? Could they not be influenced by the ongoing study ?

5/Regarding results

You need to talk about Harms. Harms that are related to the colonoscopy but also harms related to the psychological effect of screening itself.

6/Discussion:

It is also very long and messy, and the take home message is unclear

Minor revisions :

1/Misspellings in figures and Tables

2/The color red in Tables is not ideal

Author Response

Buenos días:

Estimado revisor: Adjunto en archivo pdf mis respuestas a sus recomendaciones que creo que mejoran el artículo y que podrá verificar en el manuscrito una vez que haya sido reformateado si es necesario. Espero haber realizado correctamente los cambios recomendados que podrá valorar en el manuscrito revisado y modificado. Muchas gracias por ayudarme a mejorar mi manuscrito.

Saludos cordiales 

Round 2

Reviewer 1 Report

I have reviewed the paper It is acceptable but English needs improvement

Author Response

Thank you very much for your review.

Reviewer 2 Report

Reviewing round 2:

The authors have made important changes to the manuscript and managed to address some of my comments. The English language has been improved but there are still some mistakes and some sentences were correct before editing !

Here are some points that still need to be addressed:

Major revisions:

1/First, please check that the manuscript follows the CONSORT statement of 2010 for reporting RCTs, it will help the authors to avoid forgetting important points.

The objective is clearer in the revised manuscript, but the primary outcome is not clearly stated for instance (see Methodology section).

2/Abstract:

-“GEI” should be defined in the abstract, same for “category 9”. “GEI” also appears in the main text but has never been defined.

-English language is not optimal, especially in the “Results section”

Ex: “449 polyps were visualized, including 229 adenomas of which…” ; why say “variables” plural and then only give male sex as the associated variable ?

-Conclusion is too strong : “it can be affirmed”, its better to say “our results suggest”

3.Introduction:

Last paragraph is still redundant, be more concise

Maybe address quickly in the introduction the type of screening tests that are mainly use (FOBT), rather than in the discussion only.

4/Methodology:

-Still no clear primary outcome and secondary outcomes

-Still no clear information about if the study is open labelled (which seems to be the case) or blinded, still no info on who performed the conoloscopies and who assessed the primary outcome (cf CONSORT). Some of these points are only mentioned in the discussion but should be mentioned in the methodology

5/Results

-It should be organized with titles for more clarity

Proposition:

-Characteristics of the population

-Results for the primary outcome (if its adenoma detection, it appears too far in the text)

-Results for the secondary outcomes

-Results for Harms !!! you say you record them in phase 7 but there is not even one sentence saying if there were any

-Language: Everything should be either in the past or present tense, but not mix of both. Usually past

6/Discussion:

-The adenoma detection rate is a measurement for doctors’ performance, the sentence about “patients having a high ADR” is incorrect (ref 35)

-aADR is not defined

-You can’t list limitations that you consider not being limitations! In that case they should be listed as strengths (ex: blinding, randomization…).

As an example, the population of patients is not totally representative of the entire community since only volunteers participated in the study, so it is still a limit regardless of randomization

Minor revisions

Tables and figures

Every abbreviation used in Tables or figures need to be explained in a footnote even if they appear in the main text (ex Table 1: “n” for number ; Table 2: “IG”, “CG”)

Table 1: Global BBPS was correct, and not the other way around

Table 3: p value for withdrawal time 0.0 ?

The * for significance should appear with the p values rather than OR or else

Author Response

Good Afternoon:

I am enclosing the manuscript after the recommendatios of Reviewer 2, which we have tried to respond with the best intentions because they are clarifications that improve the understanding of the article and its quality.We hope we have succeeded wih the changes kindly requested by Reviewer 2.

Thank you very much and best regards

*JA Mirón and M. Marcos como corresponding authors
